# Relationship between Green and Blue Spaces with Mental and Physical Health: A Systematic Review of Longitudinal Observational Studies

**DOI:** 10.3390/ijerph18179010

**Published:** 2021-08-26

**Authors:** Mariya Geneshka, Peter Coventry, Joana Cruz, Simon Gilbody

**Affiliations:** 1Department of Health Sciences, University of York, York YO10 4DD, UK; simon.gilbody@york.ac.uk; 2York Environmental Sustainability Institute, University of York, York YO10 4DD, UK; 3Department of Environment and Geography, University of York, York YO10 5NG, UK; joana.cruz@ucl.ac.uk; 4Population, Policy and Practice, UCL Great Ormond St. Institute of Child Health, University College London, London WC1N 1EH, UK

**Keywords:** environment, green space, blue space, mental health, long-term health, systematic review, cohort studies

## Abstract

There is growing interest in the ways natural environments influence the development and progression of long-term health conditions. Vegetation and water bodies, also known as green and blue spaces, have the potential to affect health and behaviour through the provision of aesthetic spaces for relaxation, socialisation and physical activity. While research has previously assessed how green and blue spaces affect mental and physical wellbeing, little is known about the relationship between these exposures and health outcomes over time. This systematic review summarised the published evidence from longitudinal observational studies on the relationship between exposure to green and blue space with mental and physical health in adults. Included health outcomes were common mental health conditions, severe mental health conditions and noncommunicable diseases (NCDs). An online bibliographic search of six databases was completed in July 2020. After title, abstract and full-text screening, 44 eligible studies were included in the analysis. Depression, diabetes and obesity were the health conditions most frequently studied in longitudinal relationships. The majority of exposures included indicators of green space availability and urban green space accessibility. Few studies addressed the relationship between blue space and health. The narrative synthesis pointed towards mixed evidence of a protective relationship between exposure to green space and health. There was high heterogeneity in exposure measures and adjustment for confounding between studies. Future policy and research should seek a standardised approach towards measuring green and blue space exposures and employ theoretical grounds for confounder adjustment.

## 1. Introduction

It is well established that noncommunicable diseases (NCDs) are the largest contributors to the global burden of disease [1]. NCDs are medical conditions that are non-infectious and non-transmittable from person to person, and in 2017 they accounted for 73% of all global deaths [2]. Cardiovascular disease (CVD), diabetes, cancer and chronic lung disease are the most prevalent NCDs [3] but they often tend to co-occur with common and severe mental health conditions such as depression, schizophrenia and bipolar disorder [4]. The relationship between physical and mental health is bidirectional and characterised by complex interactions [5,6]. Poor mental health increases the risk of developing NCDs due to engagement in unhealthy behaviours and low help seeking [7,8]. Having a long-term physical health condition, on the other hand, puts people at greater risk of depression and anxiety due to reduced quality of life, treatment side effects and disability [5,9]. Physical activity, diet, alcohol consumption and smoking play an important role in moderating this relationship but also independently affect the risk of developing both mental and physical health conditions [10]. While these modifiable risk factors are key drivers of NCDs, environmental exposures have also emerged as important determinants of health [11]. Noise and air pollution are now proven contributors to the global burden of disease and there is currently growing interest in studying the pathways between the natural environment and the development and progression of long-term health conditions. [11,12]. Green and blue spaces are areas of varying size that have been colonised by plants and/or fresh or saltwater. They make up a large proportion of the natural environment and can be both naturally occurring or existing as a result of human intervention [13,14]. Overall, the effects of green and blue spaces on health can be summarised by three major biopsychosocial pathways: reduction in harm (capturing and limiting air pollution, noise and heat); restoring capabilities (restoring attention and reducing stress); and building capacities (improving physical activity and social cohesion) [15,16,17,18,19,20].

There is now ample evidence about the relationship between exposure to different types of green and blue spaces and health. Cross-sectional research found greater exposure to an amount of green space and a blue space aesthetic (view from the window) to increase the odds of having good self-perceived general health [21,22]. A study on morbidity in primary care also deduced that, in general, having 10% more green space than average in the surrounding environment is associated with a lower risk of having mental and physical morbidity [23]. This relationship was stronger when green space was captured in a 1 km circular buffer than in a 3 km buffer [23]. Small reductions in CVD events, and the risk of all-cause and respiratory mortality were also observed with an increasing amount of greenness by cohort studies and meta-analysis [24,25,26]. Moreover, the size of urban green spaces affects the odds of having multimorbidity, as those with CVD and/or diabetes living near a park with a relatively small area had 3.1 times higher odds of having depression compared to those who lived near a park with a big area [27]. These relationships also vary by sociodemographic characteristics. Some studies have shown that the health benefits of green spaces are greater for those of low socioeconomic status (SES), nonwhite ethnicity and male sex [23,28,29].

Several systematic reviews of epidemiological studies have summarised the relationships between green and blue spaces and health [30,31,32,33,34,35,36]. While greater exposure to green space was associated with better mental and physical wellbeing [31], better general self-perceived health [32], reduced risk of all-cause mortality [32], reduced risk of CVD mortality, diabetes and preterm birth [33]; no relationship was observed for mental ill-health [30], cognitive functioning [34], urbanisation-related health conditions [35] and long-term physical health conditions [36]. Plausible explanations for this included poor study quality, study type or heterogeneity in exposure measurements [34,35,36]. Earlier systematic reviews studying the relationship between exposure to green space and physical long-term health conditions also found the literature to be saturated with cross-sectional studies that cannot prove causality [32,33,36].

It is apparent that a broad range of health and wellbeing outcomes have been studied in systematic reviews on green and blue space. However, the effect of the natural environment on the development of highly prevalent long-term mental and physical health conditions over time is still uncertain. This systematic review addresses several gaps in the literature. First, it captures only longitudinal observational data to study the relationship between exposures to green and blue spaces with long-term mental and physical health conditions. Longitudinal, observational studies are important in deducing causality and informing public health interventions [37]. Government bodies, such as Public Health England [38], have called for a need to improve quality, engagement and access to green spaces to promote good health, acknowledging there is high variation in the ways environmental exposures and types of health outcomes are used in research. Including both green and blue space exposure further addresses the methodological approaches in exposure measurements and aids the understanding of underlying mechanisms in the relationship. Thirdly, our systematic review aims to examine the relationship between exposure to green and blue spaces with the development and progression of multimorbidity. While prior systematic reviews have attempted to ascertain the relationship between the natural environment and single long-term conditions [33,36], little is known about the natural environment’s role in the development of multiple chronic conditions within an individual. Multimorbidity is a growing concern among aging populations because it reduces individuals’ quality of life, increases the risk of disability and puts financial strain on health systems [39]. Fourthly, the inclusion of both mental and physical health outcomes offers opportunities to identify differences in the direction and strength of associations between different outcomes.

This review, therefore, aims to:
1.Assess whether a significant relationship between exposures and outcomes exists.2.Identify the type of environmental exposures, type of health conditions and behaviours studied together in longitudinal relationships.3.Determine whether multimorbidity as a concept is studied in relation to different green/blue space exposures.

## 2. Materials and Methods

The Preferred Reporting Items for Systematic reviews and Meta-Analyses for Protocols (PRISMA-P) statement was used as guidance in protocol preparation and review reporting [40]. A protocol was registered via the International Prospective Register of Systematic Reviews (PROSPERO), identification number: CRD42020175965.

### 2.1. Selection Criteria

Studies published in academic journals in English were included. No date restrictions were applied. Only studies of a longitudinal, observational design with a population of male and/or female adults (mean population age: 18 years or older) were included. Populations with pre-existing health conditions and populations without pre-existing health conditions at baseline were included. Any study measuring green and/or blue space exposure that fits the broadly accepted definition of an area of naturally growing outdoor vegetation and/or water body was included. Studies that used objective (e.g., remote sensing) and/or subjective (self-reports) measures of green and blue spaces were eligible for inclusion. The primary outcome was mental and/or physical health. Mental health conditions included those which are classified by the National Institute for Health and Care Excellence NICE [41,42] as common (depression, generalised anxiety disorder (GAD), panic disorder, phobias, social anxiety disorder, obsessive-compulsive disorder (OCD) and post-traumatic stress disorder (PTSD)) and severe mental health disorders (bipolar disorder, psychosis and schizophrenia). As defined by the Centre for Diseases Control, physical health included NCDs with a duration of one year or more that “require ongoing medical attention or limit activities of daily living or both” [43]. Secondary outcomes related to health were also included: health-related behaviours (physical activity, diet, smoking, alcohol consumption), physical functioning, frailty and health-related quality of life (QoL). Eligible outcomes were included if they were reported via structured clinical interviews or by validated self-reported instruments.

The search strategy was compiled in consultation with an information specialist from the University of York Centre for Reviews and Dissemination. A search strategy striving for high sensitivity was run on 17 July 2020 in six online databases: Embase, GreenFILE, MEDLINE, PsycINFO, Scopus, Science Citation Index (see Appendix A). Search terms for longitudinal study design, green and blue space exposures, and mental and physical health were included and combined with appropriate Boolean operators.

### 2.2. Data Extraction

Retrieved records were imported into Rayyan, a web-based application commonly used as a screening aid. Rayyan is a validated tool for systematic review screening that allows for flexibility in setting screening standards [44,45]. After duplicates were identified and removed, study titles and abstracts were screened against the inclusion and exclusion criteria by one reviewer (MG). Following this, the full text of each potentially eligible study was screened by one reviewer (MG). Reference lists of studies were also screened for potentially eligible records. Uncertainty about the inclusion of a study at all stages of the screening process was resolved through consensus meetings with a second reviewer or an attempt to contact the authors for clarification. Relevant data from selected studies were extracted into Microsoft Excel using a prespecified data extraction form adapted from Cochrane [46,47] by the reviewers to suit longitudinal observational studies (see Appendix A). Data extraction was executed by one reviewer (MG) and accompanied by consensus meetings with a second reviewer to resolve uncertainties.

### 2.3. Quality Appraisal

The Newcastle–Ottawa Scale (NOS) was used for risk of bias assessment. It is endorsed by the Cochrane as a suitable tool for observational cohort and case-control studies [47,48] with established validity and interrater reliability [49]. NOS consists of three domains that assess the quality of the cohort study. These include *selection* of the study based on the representativeness of cohort and exposure measures; *comparability* based on the design or analysis; and *outcome assessment*, including loss and adequacy of follow-up. A star was awarded if a study met the criteria specified by NOS’ developers (See Appendix A) [48,49]. The overall rating of the study was based on the sum of the stars across all domains. Good quality was awarded if a study scored 3 or 4 stars on the selection domain and 1 or 2 stars on the comparability domain and 2 or 3 stars on the outcome domain. Fair quality was awarded if a study scored 2 stars on the selection domain and 1 or 2 stars on the comparability domain and 2 or 3 stars on the outcome domain. Poor quality was awarded to those studies that scored 0 or 1 star on the selection domain or 0 stars on the comparability domain or 0 or 1 star on the outcome domain (See Appendix A for more information) [48]. This tool allowed for selection and information bias to be assessed, particularly, sampling bias, differential loss to follow-up and confounding. One reviewer (MG) conducted the quality appraisal.

## 3. Results

### 3.1. Overview

The PRISMA-P flowchart in Figure 1 shows the process of identification, screening and inclusion of studies. The search yielded 24,176 studies after removal of duplicates (Figure 1). Of these, 23,941 were excluded during the title and abstract screening stage, leaving 233 studies for full-text assessment. One hundred and eighty-nine full-text records were excluded during that stage, leaving 44 studies for the qualitative narrative synthesis. Just under half (n = 90, 47.6%) of the excluded studies in the full-text screening stage did not include a green or blue space exposure, while another 38 (20.1%) studies did not have an observational longitudinal study design. A further 37 (19.6%) studies were excluded based on outcome, which either did not fit the definition of an NCD (n = 22), measured mortality (n = 3), did not use a validated instrument (n = 4), examined acute and/or infectious diseases (n = 7), or did not include a health condition (n = 1). Six studies were excluded because of the population type (all children) and 13 because of the publication type (one dissertation and twelve conference papers). Two records were also excluded because they were duplicates (See Appendix A).

Forty-four independent studies were included in the narrative synthesis [50,51,52,53,54,55,56,57,58,59,60,61,62,63,64,65,66,67,68,69,70,71,72,73,74,75,76,77,78,79,80,81,82,83,84,85,86,87,88,89,90,91,92,93]. The majority (n = 42) were published between 2010 and 2020 and based in high-income countries (n = 35) (Table 1). Nine studies were based in middle- and low-income countries. Study populations mainly comprised of adults aged 35 years or older (n = 31) (Table 1). Seven studies included populations of all age groups and another six included young adults (18–35 years). Most studies included both men and women participants (n = 35). Six studies included only female participants [50,65,74,75,83,84] and one study included only male participants [90] (Table 1). Almost all studies (n = 42, 95%) included predominantly healthy populations at baseline. Two studies included people with pre-existing health conditions, of which both were diabetes [53,82].

### 3.2. Quality Assessment

The methodological quality of more than half of all the included studies was rated as good (n = 24, 54.5%). Around one third (n = 14, 31.8%) of the studies scored poor and the rest (n = 6, 13.65%) scored fair on the overall NOS rating. Most studies scored high on the *comparability* domain of the scale, which assessed bias due to confounding. In general studies scored low on the *selection* and *outcome* domains (see Appendix A).

### 3.3. Exposures and Outcomes

Figure 2a,b provides an overview of the type and frequency of exposures and primary outcomes of the studies. Some studies used multiple indicators of green and blue space exposures and assessed more than one relevant outcome (see Table 1 for more information). There was high variation in exposure indicators, but a large proportion measured green space availability. The Normalised Difference Vegetation Index (NDVI) was the most frequently used indicator of green space availability, followed by percent green space. Almost all accessibility indicators measured either distance or presence of an urban park. One study measured green space usage [61], while only four studies measured exposure to blue space [68,75,79,86].

Studies examined a wide range of mental and physical health outcomes. Depression was the most frequently studied (n = 9) mental health outcome. One study examined anxiety and another schizophrenia. Ten different types of NCDs were identified, of which diabetes (n = 7), obesity (n = 6), CVD (n = 3), hypertension (n = 3), cancer (n = 3) and stroke (n = 2) were most frequently studied (Figure 2b).

### 3.4. Relationship between Exposure to Green and Blue Space and Mental and Physical Health

Table 1 presents a summary of the effect estimates for the relationship between green and blue spaces with all relevant outcomes of this review. Overall, there was mixed evidence of a relationship between exposures and outcomes. Nine studies examined whether green space affects the risk of developing depression [50,51,52,53,54,55,56,57,58] but six of those did not find a statistically significant association (n = 6) (Table 1) [51,52,53,54,57,58]. Out of those with a significant relationship, two studies found a small reduction [50,55], while one study found a small increase in the risk of depression with a greater availability of green space [56]. One study [59] found a high reduction in the risk of developing schizophrenia in those exposed to the highest quintile of NDVI compared with those exposed to the lowest quintile (HR (95% CI): 0.37 (0.25, 0.55)).

There was also mixed evidence of a relationship between exposure to green and blue space and the development of NCDs. Four studies found the risk of developing diabetes was reduced with greater exposure to an amount of green space [62,64,65,70]. The rest (n = 3) found no statistically significant relationship [63,66,69]. All studies about CVD showed a significant reduction in the risk of having CVD events with a greater exposure to green space [60,61,70]. On the other hand, only two out of six studies on the development of obesity found a statistically significant relationship [55,68]. A small reduction in the risk of developing cancer was also observed with a greater exposure to green space in one out of three studies [73].

Evidence across the retrieved studies suggests there is only a partial temporal relationship between exposure to green spaces and mental and physical health. CVD and diabetes were the conditions with strongest evidence of a protective relationship with green space. There was some evidence that the type of green space influences the relationship with health [70]. Astell-Burt and Feng [70] found exposure to a greater percent of tree canopy, but not a greater percent of total green space (tree canopy and grass cover), moderately decreased the risk of developing CVD, diabetes and hypertension. While some studies found exposure spatial scales (e.g., size of distance buffers) attenuated the relationship [55,72], in sensitivity analyses most studies found no change in effect estimates when analyses were repeated using different buffer sizes (see Appendix A). Confounding variables also varied among studies, but all adjusted for sociodemographic characteristics. Some studies additionally adjusted for environmental variables, such as season, noise, air pollution and humidity [50,58,59,62,65,67] and health behaviours, like physical activity [50,61,65,67,68,69,71,75]. No differences in relationships were observed between studies that adjusted only for sociodemographic variables and those that additionally adjusted for environmental and behavioural factors.

### 3.5. Relationship between Green Space and Physical Activity

Physical activity was the most frequently studied outcome in this review (n = 13). Over half of the studies (n = 7) measured physical activity by type, such as walking, jogging, cycling. The rest measured total physical activity over the course of a prespecified time period (Table 1). Only five studies found a significant association between green space exposure and physical activity [82,83,85,86,90]. There was some variation in adjustment for confounding variables between studies, but most adjusted for sociodemographic and neighbourhood contextual variables. Over half of the studies (n = 7) additionally adjusted for health status, including BMI, physical functioning and chronic diseases [82,84,85,87,89,90,91]. However, no patterns between confounding and statistically significant findings could be identified. While one study found differences in results between exposure buffer sizes [82], in sensitivity analyses, two studies found that the effect estimates did not change when green space was measured at different spatial scales (using different buffer sizes) [85,86].

### 3.6. Multimorbidity

This review found negligible evidence in the published literature of a longitudinal relationship between multimorbidity and green and/or blue space. One study examined how green space exposure affects the development of depression in adults with diabetes at baseline [53] and found no significant association between higher NDVI values and incident depression at the 5-year follow-up. Two studies additionally observed a general trend of improvement in frailty status with increasing greenness [80,81]. Despite being a concept closely related to multimorbidity, the studies on frailty did not conceptualise or measure multimorbidity.

## 4. Discussion

### 4.1. Relationship between the Natural Environment and Health

This systematic review showed there is currently minimal evidence of a consistent, significant longitudinal relationship between exposure to green and blue space and mental and physical health. Where statistically significant relationships existed, the associations were quite weak. Highest reductions in the risk of developing long-term health conditions with greater exposure to green space was observed for diabetes, CVD, stroke and schizophrenia. While prior systematic reviews and observational studies have shown there to be some significant cross-sectional associations between depression, diabetes and obesity [33,36,94,95,96], this systematic review concludes the relationship does not generally hold longitudinally. Due to the recent nature of the research, the reasons behind this are not entirely clear. One potential explanation could be the methodological design of longitudinal studies and the measurement of environmental exposures. First, the heterogeneity of green space exposure measures is well documented in the academic literature [34,97,98]. This is also supported by studies in our systematic review. A range of data sources, including remote sensed imagery from land use maps, regional government databases and self-reported information, is commonly used to ascertain green space exposure in the neighbourhood [99]. Such data sources are often incomplete and provide a varying degree of accuracy, which increases the difficulty of sourcing enough data to measure green space both at baseline and follow-up. Very often, green space exposures in longitudinal studies are measured only at one point in time with the assumption that the presence of vegetation doesn’t change drastically over time [51,53,54,58,59,61,63,65,66,70,71,74,76,80,83,84,85,86,87,88,89,91]. However, urban areas undergoing regeneration or expansion may experience drastic changes in the amount and availability of greenery [100]. While cross-sectional studies only measure green space at a single point in time, longitudinal studies require multiple and complex exposure measurements. The unavailability of data to assess these changes in exposure over time could be a reason for the lack of longitudinal relationships.

Another potential explanation for the differences in results between cross-sectional and longitudinal studies could be the duration of follow-up of longitudinal studies. The dosage and duration of green space exposures required to influence health is still not entirely understood. However, there is some evidence that environmental factors in childhood and even from preconception and birth can shape the health of a person decades later [101]. Sensitive periods during human development are discrete time points at which certain environmental stimuli must be encountered for mental and physical development to occur [102]. The need to incorporate a life-course approach when studying the effects of green spaces on health has been previously highlighted, but its feasibility requires extensive utilisation and interpolation of historical data from varying sources [103]. While positive associations between green space and health observed in cross-sectional studies may be caused by sample size or sampling bias, the lack of relationship at a longitudinal level may be due to the low duration of follow-up. More research, therefore, is required to understand whether exposure to green space during sensitive periods of human development affects health later in life. This would better inform the duration of follow-up and study design of future longitudinal research.

It should be noted that our systematic review examined a broad range of mental and physical health outcomes, which yielded different strengths of associations. A finding that stood out was the relationship between exposure to green space and schizophrenia [59]. Chang et al. [59] found the risk of developing schizophrenia to be reduced by 63% (HR (95% CI): 0.37 (0.25, 0.55)) in those exposed to the highest quintile NVDI compared to those exposed to the lowest. This is consistent with prior research on the relationship between green space and schizophrenia [104]. The reasoning behind these findings is not entirely clear but it is known that the risk of schizophrenia is often influenced by environmental exposures such as air pollution and urbanicity [105]. Biological mechanisms that affect brain development is a potential explanation for the increased risk of developing schizophrenia with greater exposure to air pollution [105]. As green spaces have the ability to reduce and capture air pollution, it is plausible that they counteract the negative effects of hazardous environmental factors.

Confounding could be a potential contributor to differences in results between studies included in this systematic review. Variation in confounding between studies was observed, but most adjusted for sociodemographic variables, such as age, sex and socioeconomic position. Although some studies additionally adjusted for physical activity, air quality and noise, no differences in relationships could be observed between minimally adjusted studies and those adjusting for additional environmental and behavioural variables. The review deduced there is currently no consensus on appropriate confounder adjustment, but it should be acknowledged that additional contextual factors like the built environment and clinical characteristics can also have an impact on the relationship. For example, studies have shown that neighbourhoods with high crime, deprivation, social disorganisation, a high retail density and land-use mix, can increase the risk of depression [106,107]. It is also hypothesised that further consideration of childcare duties and types of work might play an important role in the ways people utilise and interact with their environment [108]. We found that studies in this systematic review generally lacked adjustment for such variables, possibly due to a lack of such data in health cohorts.

Apart from confounding, differences in results could be due to exposure measurements. This review found a broad range of exposure indicators were used to conceptualise green space. The NDVI, percent green space and distance to park were the most frequently used, however, there was high heterogeneity between studies on the choice of spatial scale and exposure classes. Buffer sizes, time-of-year NDVI measurements and other green space exposure data sources varied, making meaningful comparisons between studies difficult and a potential reason for the differences in results. These findings have been previously flagged in prior systematic reviews [34,109,110]. Where studies examined the type of green space, they mainly included urban parks. For most, this was measured as either the distance from the residential address or presence within a distance buffer. These are common measures of green space accessibility [111] but have some limitations. First, such spatial measures fail to capture specific characteristics and features of urban parks. Some research, for example, indicates that physical activity is higher in parks with paved trails [112], and visits to green spaces are more likely to occur if they have certain attributes, like trees, toilets, gym facilities, and the presence of lakes, ponds and trees [113,114]. Only one study included in this systematic review conducted a comparative analysis between exposure to trees and the total amount of vegetation in the neighbourhood [70]. They found the risk of CVD, diabetes and hypertension were all reduced with greater exposure to percent tree canopy cover, but not with greater exposure to percent total green space [70]. Greater exposure to street trees has been previously shown to reduce the odds of having hypertension [115] and poor mental health [116]. While other studies of this review compared effect estimates using different buffer sizes (and found negligible differences), this finding suggests that it is the type and location of green spaces rather than the spatial scale that affects health. However, further comparative research is needed to establish this.

### 4.2. Strengths and Limitations

To the best of our knowledge, this is the first systematic review to summarise the published longitudinal literature on the relationship between green and blue spaces and chronic health. This is important for informing intervention design and policy decision making. According to the Medical Research Council’s framework for evaluating complex interventions [117], appropriate methods need to be employed to first identify existing evidence and use it to guide theory development that is critical to intervention design. This systematic review contributed to the identification and synthesis of existing evidence and could help bridge the gap between empirical research and the development of programme theory about the role of green space in the maintenance of mental and physical health. Including both mental and physical health outcomes as well as related health states and behaviours additionally allowed for a comprehensive analysis and summary of the effects of the natural environment on highly prevalent NCDs and mental health problems. It also enabled comparisons of the strength and direction of associations. The choice to include both green and blue spaces as exposures, on the other hand, better informed of current research gaps in the published literature on the relationships between water bodies and health. Lastly, we summarised the limited evidence of longitudinal relationships between green and blue spaces and multimorbidity. While prior systematic reviews have assessed the effects of green spaces on health, they have not considered how these exposures may influence the development of multiple chronic conditions within an individual [30,31,32,33,34,35,36]. This systematic review, therefore, flags additional research gaps in the study of multimorbidity development in relation to the natural environment.

There are a number of limitations. First, heterogeneity in study exposures and populations prevented us from conducting a quantitative synthesis analysis. While a narrative synthesis enabled a summary of results and associations, a meta-analysis may improve generalisability of the results by producing a pooled effect estimate and identifying sources of heterogeneity and bias [118]. Second, the Newcastle–Ottawa Scale is not as robust and as comprehensive a measure as ROBINS-I which is widely regarded as offering gold standard assessments of risk of bias of nonrandomised intervention studies [119]. The exposure domain on the Newcastle–Ottawa Scale might not be optimal for assessing information bias because it only classifies the quality of a study as good if the exposure is measured through objective measures. In the context of our review, objective measurements of green space are typically made by professional assessments or satellite imagery. However, self-reported exposures of natural environments are important in assessing the ways people interact with these spaces and may not necessarily introduce recall bias like clinical exposures [120]. Additionally, the Newcastle–Ottawa Scale includes domains that are critical to assessing key parameters of methodological quality of longitudinal cohort studies and in this sense functioned as a pragmatic solution for this review.

### 4.3. Review Implications

Despite the qualitative analysis of this review showing little relationship of exposure to green and blue space with health, this systematic review aided the identification of some key research gaps. First, there is a lack of framework to study the type and components of green and blue spaces on health. Longitudinal research has typically used an average estimation of green space availability or accessibility, and this is loosely based on European Environment Agency [121] and Natural England’s [122] recommendations of having an accessible green area of at least 2 ha no more than 300 m or within a 15-min walk from the residential address. Future research, however, should adopt a more holistic approach whereby different characteristics, dosage of exposure and specific person–environment interactions are studied in relation to health. This could improve the understanding of the different pathways between green space exposure and health, and lead to the design and implementation of evidence-based public health interventions.

Second, there is a need for more research into the relationship between blue space and health, as only four longitudinal studies were identified [68,75,79,86]. Prior academic literature has conceptualised the relationship between blue space and health to be driven by socio-environmental factors similar to those for green space [123]. Unlike green space, health policy recommendations for accessibility or availability of blue space are limited and primarily focused on coastal zones [124]. Government bodies and environmental agencies, therefore, should seek to develop more robust guidelines based on empirical research.

Finally, this review identified a lack of research into the ways green and blue spaces affect the development of multiple chronic conditions within an individual, also known as multimorbidity. The management of multimorbidity usually requires complex clinical interventions that have a negative impact on quality of life and put strain on healthcare systems [125,126,127]. Green and blue spaces can influence behavioural change and promote good health through socio-ecological pathways and so the natural environment could play an important role in reducing the multimorbidity burden by preventing the onset or slowing the progression of several chronic conditions.

## 5. Conclusions

This systematic review showed there to be mixed evidence of a longitudinal relationship between green and blue spaces and mental and physical health, with just over half of all analyses indicating a nonsignificant relationship between exposures and health outcomes. The majority of published longitudinal observational studies assess exposure to green space through indicators of availability or urban green space accessibility. Few studies assess the effects of blue spaces on health. There was high heterogeneity between studies in exposure measures and confounding. This could be explained by a lack of existing framework and uniform guidelines on studying the effects of the natural environment on health. Future longitudinal research should incorporate a more holistic approach towards conceptualising green and blue space that moves beyond the amount or distance and towards capturing types and characteristics. This could greatly aid the understanding of causal pathways and improve intervention design.

## Figures and Tables

**Figure 1 ijerph-18-09010-f001:**
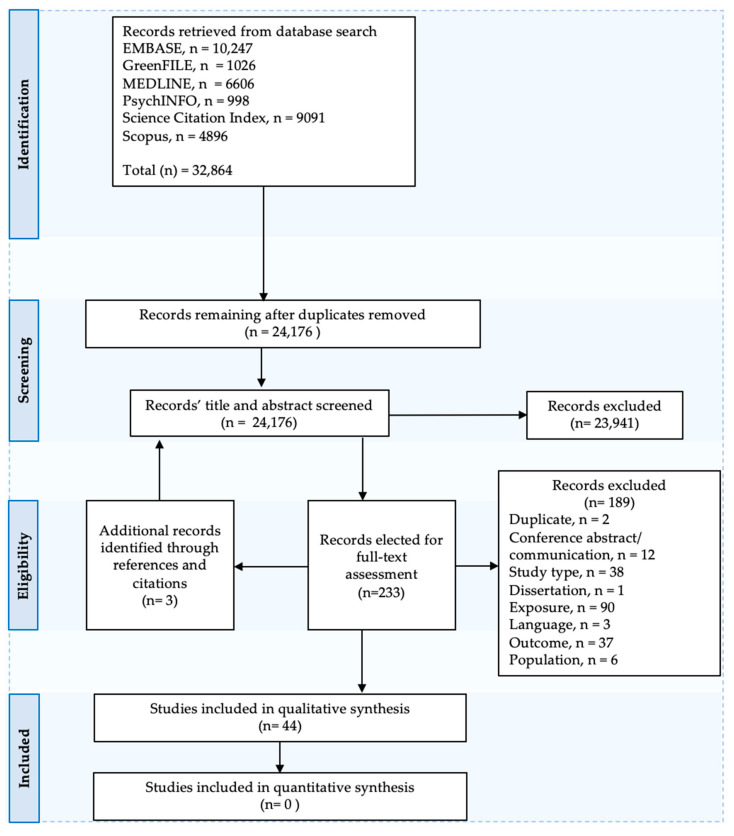
PRISMA-P Flowchart.

**Figure 2 ijerph-18-09010-f002:**
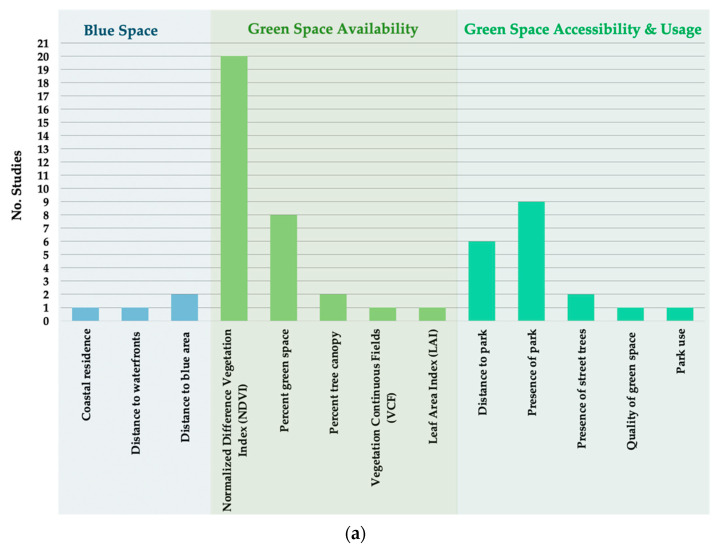
Frequency and type of selected studies by: (**a**) exposure; (**b**) primary outcome.

**Table 1 ijerph-18-09010-t001:** Summary of study characteristics, results and quality appraisal.

Study Reference	Population Description	Sample Size	Cohort Name/Data Source	Follow-Up Duration	Exposure Indicator Description	Exposure Indicator Type	Outcome	Outcome Measure	Main Results Effect Estimate [95% CI ^1^]	Confounders	Study Quality *
Primary Outcomes											
Mental Health											
Banay et al. [50]	women nurses; ≥30–55 years [USA]	121,701	Nurses’ Health Study	10 years	NDVI ^1^ averages for each year of follow-up; 250 m and 1250 m circular buffers	Availability	Depression	First self-report of physician/clinician diagnosis of depression or new regular use of antidepressants	250m Buffer HR ^1^: 0.87 [0.78, 0.98] Highest NDVI quintile 1250 m Buffer HR: 0.90 [0.80, 1.02] Highest NDVI quintile	age, race, mental health, marital status, educational attainment, husband’s educational attainment, population density, income, median home value, PM ^1^ 2:5 level, BMI ^1^, smoking status and pack-years of smoking, alcohol consumption, physical activity, physical function, bodily pain [baseline], social network strength, care to ill family members [baseline], difficulty sleeping [baseline]	Poor
Fernandez-Nino et al. [51]	men and women; ≥55 years [Mexico]	1524	Study on Global Ageing and Adult Health [SAGE]	5 years	Street trees; total length of street covered in trees in a 950 m road network buffer	Accessibility	Depression	Self-report of physician diagnosis	OR ^1^: 0.90 [0.29, 2.83] Highest quintile of street length covered in trees	sex, age, income index, functional limitations, margination index of the municipality	Good
Gariepy et al. [52]	men and women; ≥18–80 years [Canada]	13,618	National Population Health Survey	10 years	Presence of a park within a 500 m circular buffer	Accessibility	Depression	Self-reported instrument	Β ^1^: −0.4 [−1.4, 0.6] For answering “yes” to presence of a park	age, sex, marital status, education, income adequacy, childhood life events, chronic condition, family history of depression	Good
Gariepy et al. [53]	men and women; ≥18–80 years; with diabetes [any type] [Canada]	2003	Diabetes Health Study [DHS]	5 years	NDVI	Availability	Depression	Self-reported instrument	HR: 0.94 [0.88, 1.01] Per decile increase in NDVI	sex, age, marital status, family income, educational level, employment	Good
Melis et al. [54]	men and women; ≥20–65 years [Italy]	547,263	Turin Longitudinal Study [TLS]	2 years	Availability of green space measured via index by area units	Availability	Depression	Antidepressant use	Men IRR ^1^: 0.98 [0.92, 1.04] Highest index value quintile green Women IRR: 1.00 [0.96, 1.08] Highest index value quintile of green	sex, age, education level, activity status, citizenship, residential stability at same address	Good
Picavet et al. [55]	men and women; ≥18 to 55 years [Netherlands]	4917	Doetinchem Cohort Study	15 years	Percent green space in 125 m and 1000 m circular buffer	Availability	Depression	Self-reported instruments	Per unit increase in percent green space 125 m OR: 0.97 [0.92, 1.04] 1000 m OR: 0.86 [0.79; 0.93]	age, sex, SES ^1^	Poor
Tomita et al. [56]	men and women; mean 20 years [South Africa]	11,156	South African National Income Dynamics Study [SA-NIDS	4 years	NDVI, 250 m resolution square	Availability	Depression	Self-reported instrument	OR: 1.01 [1.01, 1.02] Each unit increase in NDVI value	age, sex, marital status, race, household income, employment, rurality	Good
Astell-Burt and Feng [57]	men and women; ≥45 years [Australia]	46,786	45 and Up Study	6.2 [mean] years	Total percent green space; tree canopy in a 1600 m road network buffer	Availability	Depression or anxiety	Self-report of doctor diagnosed	OR: 1.26 [0.89, 1.63] Highest percent quintile total green OR: 0.86 [0.80, 1.01] Highest percent quintile tree canopy	age, sex income, education, economic status, couple status	Poor
Pun et al. [58]	men and women; ≥57–85 years [USA]	3005	National Social Life, Health, and Aging Project [NSHAP]	6 years	NDVI seasonal changes in 1000 m circular buffer	Availability	Depression; anxiety	Self-reported instrument	Anxiety β: −0.104 [−0.322, 0.115] per unit increase in NDVI Depression β: −0.274 [−0.596, 0.048] per unit increase in NDVI	age, gender, questionnaire year, season, region, education attainment, 3-day moving average of temperature, 60-months moving average of PM2.5	Good
Chang et al. [59]	men and women mean age: 43.36 [20.44] years [Taiwan]	869,484	Taiwan Longitudinal Health Insurance Database	10 years	NDVI at baseline; 2000 m circular buffer around hospital most frequently visited	Availability	Schizophrenia	Physician-diagnosed	HR: 0.37 [0.25, 0.55] Highest NDVI quintile	age, sex, health insurance rate, classification of the insured, temperature, relative humidity, precipitation	Good
NCDs											
Dalton and Jones [60]	men and women; mean 59.2 years [United Kingdom]	25,639	European Prospective Investigation of Cancer [EPIC] Norfolk	14.5 [mean] years	Percent green space in 800 m circular buffer	Availability	CVD ^1^	Health register	HR: 0.93 [ 0.88, 0.97] Highest percent quintile green	sex, age, BMI, diabetes, SES [individual and neighbourhood]	Good
Tamosiunas et al. [61]	men and women; ≥45–72 years [Lithuania]	5112	Health, Alcohol, and Psychosocial Factors in Eastern Europe [ HAPIEE]	4.41 [mean] years	Distance to park and park use [self-reported]	Accessibility	CVD	Self-reported doctor diagnosed	User: HR: 1.58 [0.95, 2.63] Longest distance quintile Nonuser: HR: 1.66 [1.01, 2.73] Longest distance quintile	age, sex, education, smoking, arterial hypertension, physical activity, total cholesterol level, fasting glucose level, BMI, diabetes mellitus, cognitive function, symptoms of depression, self-rated health, and quality of life	Good
Clark et al. [62]	men and women; ≥45–84 years; urban residents [Canada]	380,738	British Columbia mandatory health insurance database	4 years	NDVI yearly and seasonal; in 100 m circular buffer	Availability	Diabetes	Health register	OR: 0.90 [0.87, 0.92] IQR ^1^ increase in NDVI	sex, age, area-level household income, walkability, pollution	Good
Renzi et al. [63]	men and women; ≥35 years [Italy]	1,459,671	Rome Longitudinal Study	5.2 [mean] years	NDVI and LAI in a 300 m circular buffer	Availability	Diabetes	Medical records	β: −1.87 [−7.40, 3.99] Per unit increase in NDVI	SES, marital status, educational level, occupation, place of birth, sex	Good
Dalton et al. [64]	men and women; ≥40–80 years [United Kingdom]	25,633	European Prospective Investigation into Cancer [EPIC] Norfolk	11.3 [mean] years	Percent green space; in 800m circular buffer	Availability	Diabetes[T2]	Self-report of physician diagnosis or medication	HR: 0.81 [0.65, 0.99] Highest percent quintile green	sex, age, BMI, parental diabetes, SES	Good
Liao et al. [65]	pregnant women; 25–29 years mean age group [China]	6,883	Visitors of Wuhan’s Women and Children Medical and Healthcare Center	9 months or until development of gestational diabetes	NDVI for conception years; 300 m circular buffer	Availability	Diabetes [genstational]	Clinical samples	RR^1^: 0.66 [0.52, 0.84] Highest quintile NDVI	age, education years, BMI, passive smoking during pregnancy, parity, season	Good
Hobbs et al. [66]	men and women; ≥18–89 years [United Kingdom]	28,806	Yorkshire Health Study	3 years	Presence of park in a 2000 m circular buffer	Accessibility	Obesity	BMI, self-report	OR: 0.99 [0.98, 1.02] for answering “*yes*” to presence of park	age, sex, education, deprivation, population density	Fair
Persson et al. [67]	men and women, ≥35–65 years [Sweden]	5712	Stockholm Diabetes Prevention Program [SDPP]	8.9 [mean] years	NDVI; time-weighted in a 100 m, 250 m, 500 m circular buffer	Availability	Obesity	Objective measures of BMI	IRR for IQR increase in NDVI 500 m Females: 1.05 [0.88, 1.26] Males: 1.06 [0.89, 1.26]	age, alcohol consumption, tobacco use, psychological distress, shift work, aircraft noise, railway noise, distance to water	Good
Halonen et al. [68]	men and women; public sector employees; mean: 47.7 years [nonmovers] and among the movers 41.8 [Finland]	35,213	Finnish Public Sector study	8 years	Distance to green space; distance to blue space in meters, objectively measured	Accessibility	Obesity and overweight	Self-reported BMI	Green space OR: 1.50 [1.07, 2.11] Longest distance quintile Blue space OR: 1.15 [0.94, 1.39] Longest distance quintile	age, sex, education, chronic disease, neighbourhood socioeconomic disadvantage, BMI, smoking, heavy alcohol, physical inactivity	Poor
Lee et al. [69]	men and women; ≥19 years [48.6 years mean] [USA]	5435	Offspring and Generation Three Cohorts of the Framingham Heart Study	6.4 years	Percent green space within a census block	Availability	Obesity; Diabetes	Blood samples; medication; objectively-measured BMI	Diabetes: OR: 0.70 [0.41, 1.19] Highest percent quintile green Obesity: no results	age, gender, smoking status, education, cohort status, fasting plasma glucose, BMI	Fair
Astell-Burt and Feng [70]	men and women; ≥45 years [Australia]	53,196	45 and Up Study	6 years	Percent green space; tree canopy in a 1600 m road network buffer	Availability	Diabetes, hypertension and CVD	Self-report of physician diagnosis	Diabetes OR: 1.10 [0.65, 1.95] Highest percent quintile total green OR: 0.71 [0.56, 0.91] Highest percent quintile tree canopy Hypertension OR: 0.72 [0.64, 1.12] Highest percent quintile total green OR: 0.82 [0.71, 0.95] Highest percent quintile tree canopy CVD OR: 0.89 [0.59, 1.13] Highest percent quintile total green OR: 0.79 [0.63, 0.92] Highest quintile tree canopy	age, sex income, education, economic status, couple status	Poor
Paquet et al. [71]	men and women; ≥18 years [Australia]	4056	North West Adelaide Health Study [NWAHS]	3.5 [mean] years	NDVI in 1000 m road network buffer	Availability	Diabetes; hypertension; obesity; dyslipidaemia	Clinical samples	Per unit increase in NDVI Diabetes RR: 1.01 [0.90, 1.13] Hypertension RR: 0.97 [0.87, 1.07] Dyslipidaemia RR: 1.12 [1.00, 1.25] Obesity RR: 1.04 [0.92, 1.16]	age, gender, smoking status, education, cohort status, fasting plasma glucose, BMI	Good
de Keijzer et al. [72]	men and women; ≥35–55 years civil servants [United Kingdom]	10,308	Whitehall II	14.1 [median] years	NDVI and VCF, 500 m and 1000 m circular buffers and LSOA	Availability	Metabolic Syndrome	Clinical samples	IQR increase in NDVI 500m HR: 0.87 [0.77, 0.99] 1000 m HR: 0.90 [0.79, 1.01] LSOA HR: 0.91 [0.79, 1.03]	age, sex, ethnicity, individual socioeconomic status [education and employment grade], neighbourhood socioeconomic status [income and employment deprivation]	Good
Datzman et al. [73]	men and women; mean 49.33 years; [Germany]	1,918,449	AOK Plus [health insurance database]	4 years	NDVI; 115 images for 4 years; statistical area units	Availability	Cancer: colorectal; mouth and throat, prostate, breast; nonmelanoma skin	Health register	Per 10% increase in NDVI Colorectal: RR: 1.03 [0.98, 1.07] Mouth: RR: 0.89 [0.83, 0.96] Skin: RR: 0.84 [0.79, 0.90] Prostate: RR: 0.95 [0.90, 1.01] Breast: RR: 0.96 [0.92, 0.99]	age, sex, alcohol-related disorder, absolute number of physician contacts, proportion of short and long-term unemployment	Good
Conroy et al. [74]	women; ≥45–75 years; [African Americans, Japanese Americans, Latinos, Native Hawaiians, and White] [USA]	48,247	Multiethnic Cohort [MEC]	17 years	Presence of a park; based on number in a residential block group	Accessibility	Breast cancer [invasive]	Health register	HR: 1.03 [0.92, 1.15] No park in area	age, clustering effect of block group, ethnicity, risk factors, baseline BMI and adult weight change, neighbourhood SES, all neighbourhood obesogenic factors	Good
Haraldsdottir et al. [75]	women; mean: 53.9 years [Iceland]	10,049	Reykjavik Study	27.3 average	Coastal residence, self-reported	Availability	Breast cancer	Health registers	HR: 0.87 [0.72, 1.04] Coastal residence vs. city	age, birth cohort, education, physical activity, parity, height, BMI in midlife, age at menarche, age at first child	Good
Orioli et al. [76]	men and women; ≥30 years [Italy]	1,265,058	Rome Longitudinal Study	13 years	NDVI average for 2015 in 300 m and 1000 m circular buffer	Availability	Stroke	Health register	NDVI highest quintile 300 m HR: 0.95 [0.91, 0.98] 1000 m HR: 0.97 [0.93, 1.00]	age, sex, educational level, marital status, occupational status, place of birth, area-level SES	Good
Paul et al. [77]	men and women; ≥35–100 years; urban residents Ontario [Canada]	4,251,146	Ontario Population Health and Environment Cohort [ONPHEC]	13 years	NDVI annual values, 250 m circular buffer	Availability	Stroke	Health register	HR: 0.96 [95% CI: 0.95, 0.97] per IQR increase in NDVI	age, sex, SES, comorbidities, northern residence, population density, air pollution	Good
Yuchi et al. [78]	men and women; ≥45–84 years [Canada]	634,432 [parkinson disease]; 7232 [multimple sclerosis]	Medical Services Plan [MSP] Vancouver, mandatory health insurance database	4 years	NDVI; yearly average in 100 m circular buffer	Availability	Parkinson’s disease Multiple sclerosis	Health records	Per IQR increase in NDVI Parkinson’s Disease: OR: 0.97 [0.93, 1.01] Multiple Sclerosis: OR: 1.14 [1.00, 1.30]	Parkinson’s disease: age, sex, comorbidities, household income, education, ethnicity Multiple sclerosis: age, sex, comorbidities, household income, education and ethnicity, comorbidities, household income, education, ethnicity	Good
Picavet et al. [55]	men and women; ≥18 to 55 years [Netherlands]	4,917	Doetinchem Cohort Study	15 years	Percent green space in 125 m and 1000 m circular buffer	Availability	Obesity; Hypertension	All self-reported instruments	Per unit increase in percent green space 125 m Obesity: OR: 1.04 [1.01, 1.07] Hypertension: OR: 0.99 [0.97, 1.02] 1000 m Obesity: OR: 1.00 [0.96; 1.05] Hypertension: OR: 1.02 [0.98; 1.05]	age, sex, SES	Poor
Secondary Outcomes											
de Keijzer et al. [79]	men and women; ≥35–55 civil servants [United Kingdom]	10,308	Whitehall II study	9 [median] years	NDVI and EVI; distance to blue space [any visible water]; distance to green or blue space in 500m and 1000 m circular buffer; distance in m	Availability Accessibility	Physical Functioning	Clinical measures	Walking speed [difference baseline and follow-up]: 500 m NDVI β: 0.02 [0.01, 0.04] per IQR increase 1000m NDVI β: 0.03 [0.01, 0.04] per IQR increase Blue space β: −0.01 [−0.02, 0.01] per IQR increase Grip strength [difference baseline and follow-up]: 500 m NDVI β: −0.01 [−0.03, 0.01] per IQR increase 1000 m NDVI β: −0.01 [−0.03, 0.01] per IQR increase Blue space β: −0.01 [−0.03, 0.01] per IQR increase	sex, ethnicity, marital status, height, alcohol use, intake of fruit and vegetables, smoking, rurality, education, employment grade, Index of Multiple Deprivation [IMD], income score and of the IMD, employment score	Fair
Yu et al. [80]	men and women; ≥65 years [Hong Kong]	4000	Mr and Ms Os Study	2 years	NDVI at baseline in a 300 m circular buffer	Availability	Frailty	Self-reported instrument	OR: 1.29 [1.04, 1.60] Highest quintile NDVI	age, sex, marital status, SES, current smoking status, alcohol intake, diet quality, baseline frailty status, number of diseases, cognitive function, physical activity, depression	Good
Zhu et al. [81]	men and women; ≥65 years [China]	34,342	Chinese Longitudinal Healthy Longevity Survey [CLHLS]	9 years	NDVI; annual averages for each year in 500 m buffer	Availability	Frailty	Self-reported instrument	OR: 1.02 [1.00, 1.04] Per unit increase in NDVI	age, sex, ethnicity, marital status, geographic region, urban or rural residence, education, occupation, financial support, social and leisure activity, smoking status, drinking status, physical activity	Good
Chong et al. [82]	men and women; ≥45 years with diabetes [T2] [Australia]	60,404	45 and Up Study and the follow-up Social, Economic and Environmental Factors [SEEF] Study	3.3 [mean] years	Percent green space in 500 m, 1000 m, and 2000 m road network buffer	Availability	Physical Activity	Self-reported instrument [MVPA: min/week]	Per highest percent quintile green 500 mMean: 0.61 [−0.26, 1.49] 1000 mMean: 0.94 [0.10, 1.79] 2000 mMean: 0.75 [0.03, 1.48]	age, sex, country of birth, education, disadvantage, physical functioning, BMI, psychological distress	Poor
Cleland et al. [83]	women parents; mean: 42.4 years;[Australia]	698	Children Living in Active Neighbourhoods [CLAN]	2 years	Amount of greenery and quality of parks, self-reported satisfaction	Availability Accessibility	Physical activity	Self-reported instrument [walking: for leisure and transport [min/ week]]	Amount of greeneryPersistently high vs. persistently low PA: RR: 1.80 [1.04, 3.13] Increased vs. persistently low PA: RR: 1.39 [0.90, 2.17] Quality of parks Persistently high vs. persistently low PA: RR: 1.73 [1.17, 2.57] Increased vs. persistently low PA: RR: 1.20 [0.89, 1.62]	age, marital status, number of children in the household, highest level of schooling	Poor
Coogan et al. [84]	women; ≥21–69 years; Black ethnicity [USA]	21,820	Black Women’s Health Study	2-6 years 98,280 person-years of follow-up.	Distance to park	Accessibility	Physical activity	Self-reported instrument [Walking for recreation and total walking: y/n]]	Recreation walking OR: 1.01 [0.89, 1.13] Shortest distance quintile Exercise walking OR: 1.01 [0.91, 1.12] Shortest distance quintile	age, region, BMI, smoking, alcohol, marital status, parity, caregiver status, residential moves, chronic conditions, history of cancer, moving residence, vacant housing, SES, crime	Poor
Dalton et al. [85]	men and women; mean age at baseline 62.2 [United Kingdom]	25,639	European Prospective Investigation into Cancer [EPIC] Norfolk	7.5 [mean] years	Percent green space at baseline for nonmovers; 800 m	Availability	Physical Activity	Self-reported instrument [Change in overall PA [hr/week]]	β: 4.21 [1.60, 6.81] Highest percent quintile green	age, sex, marital status, waist to hip ratio, BMI, morbidity, urban/ rural location	Fair
Faerstein et al. [86]	men and women; ≥18 years; civil servants [Brazil]	1731	Pro-Saude study	13 years	NDVI [800 m circular buffer]; presence of trees [visual inspection]; proximity to waterfronts;	Availability Accessibility	Physical activity	Self-reported instrument [nonwork PA: yes/no]	OR: 0.85 [0.44, 1.65] Highest quintile NDVI OR: 1.22 [0.62, 2.40] Highest percent quintile of trees OR: 2.46 [1.22, 4.93] Longest distance to waterfronts	sex, race, education, income, neighbourhood contextual variables	Poor
Hogendorf et al. [87]	men and women; mean: 53 years; [Netherlands]	4758	Gezondheid en Levens Omstandigheden Bevolking Eindhoven en omstreken [GLOBE]	10 years	Area of green space within a 1000 m circular buffer; Distance to green space	Availability Accessibility	Physical activity	Self-reported instrument [total walking and cycling: min/week]	Total walking and cycling Per ha increase in area of green β: 0.82 [−178.84, 180.48] Distance per 100m increase in green β: −22.36 [−46.19, 1.48]	marital status, income, employment, smoking, self-rated health	Poor
Josey and Moore. [88]	men and women; ≥25years; urban residents [Canada]	2707	Montreal Neighborhood Networks and Healthy Aging Panel [MoNNET-HA]	5 years	Distance to parks and green spaces	Accessibility	Physical Activity	Self-reported instrument [physical inactivity: y/n]	OR: 0.99 [0.99, 1.00] Per mile increase in distance	sex, age, self-reported health status, SES, household language, marriage status, residential duration, wave	Poor
Lin et al. [89]	men and women; ≥65–98 years [Hong Kong]	4000	OS and Ms. OS Study	7.8 [mean] years	NDVI in 300 m circular buffer	Availability	Physical activity	Self-reported instrument [Total PA score]	No relevant results	age, sex, marital status, education level, alcohol consumption, smoking, living alone, self-rated health, chronic conditions, functional impairment	Fair
Michael et al. [90]	men; ≥65 years [USA]	513	Neighborhoods and Physical Activity in Elderly Men	3.6 [mean] years	Distance to park	Accessibility	Physical activity	Self-reported instrument [walking: min/day]	RR for presence of park Low SES: 0.89 [0.70, 1.13] High SES: 1.34 [1.16, 1.55]	age, race education, occupation, marital status, self-reported health, BMI, smoking, drinking, chronic conditions	Fair
Sugiyama et al. [91]	men and women; mean: 54.4 years [Australia]	4802	AusDiab study	7 years	Park or nature reserve in the neighbourhood, self-reported	Accessibility	Physical Activity	Self-reported instrument [meeting PA guidelines: y/n]	OR: 0.96 [0.80, 1.15] for having a park in neighbourhood	age, sex, education, work status change, child change, mobility, BMI	Poor
Yang et al. [92]	men and women; ≥40–79 years [United Kingdom]	25,633	European Prospective Investigation into Cancer [EPIC] Norfolk	7 years	Presence of park or green space in 800 m circular buffer	Accessibility	Physical activity	Self-reported instrument [active commuting: y/n]	Park [yes]: OR: 1.30 [0.96, 1.74] Green space [yes]: OR: 1.12 [0.83, 1.53]	No adjustment	Poor
Meyer et al. [93]	men and women; ≥18–30 years; black and white [USA]	5115	Coronary Artery Risk Development in Young Adults [CARDIA]	13 years	Number of parks within a 3000 m circular buffer	Accessibility	Physical activity; Diet Quality	Self-reported validated instruments [PA: frequency walking, biking, running/ week]	No relevant results	N/A	Poor
Picavet et al. [55]	men and women; ≥18 to 55 years [Netherlands]	4917	Doetinchem Cohort Study	15 years	Percent green space in 125 m and 1000 m circular buffer	Availability	Physical activity; Quality of Life	All self-reported instruments [PA: meeting guidelines: y/n]	Per unit increase in NDVI 125 m Physical activity: OR: 1.02 [0.99; 1.04] Quality of Life: Mixed 1000 m Physical activity: OR: 1.01 [0.97; 1.05] Quality of Life: Mixed	age, sex, SES	Poor

^1^ Abbreviations: BMI: Body Mass Index/CI: Confidence Intervals/HR: Hazard Ratio/IQR: Interquartile Range/MVPA: Moderate-to-vigorous physical activity/NDVI: Normalized Difference Vegetation Index/OR: Odds Ratio/PA: Physical activity/PM: Particulate matter/RR: Relative Risk/SES: Socioeconomic status/β: Beta coefficient; * Based on Newcastle–Ottawa Scale [NOS] for Cohort Studies.

## Data Availability

The data presented in this study are available on request from the corresponding author. All data used in the production of this review are included in the published studies.

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
