# Peer review of "Relationship between Green and Blue Spaces with Mental and Physical Health: A Systematic Review of Longitudinal Observational Studies"

_ijerph, 2021, doi:10.3390/ijerph18179010_

Round 1

Reviewer 1 Report

I would like to thank for the opportunity to review this paper.

The manuscript addresses one interesting topic and is very well structured. 

Some revisions of the English should be made.

ABSTRACT

The authors should clarify the sentence “Common and serious mental health conditions as well as non-communicable diseases (NCDs)”. There are several NCDs that are common and serious. The conjunction “as well as” should not be used here

Please clarify how a search in 6 databases only retrieved 44 eligible studies. I think the authors mean that the 44 studies were assessed eligible after screening and not immediately after the search.

If no meta-analysis was performed, how can the authors say: “There was mixed evidence of a statistically significant relationship across most outcomes.”?

Please clarify:  “This could be attributed to differences in exposure measures and adjustment for confounding.”

INTRODUCTION

Please revise the classification used: “common mental health problems such as depression and serious mental illnesses“. Depression is also a serious mental illness. Maybe use DSM, for example.

The relevance and innovation of the study should be better addressed in the Introduction.

METHODS

Lines 120-121: How was this definition applied to the included studies?

2.4. Quality Appraisal 

The Newcastle-Ottawa Scale has been subject of critics. Please see: https://www.bmj.com/content/355/bmj.i4919

Please indicate whether the quality appraisal was done by one or two researchers.

RESULTS

Line 195: “The majority of exposures were associated with green space (n=40)”. The sum of the nº of studies in the figure with green as exposure does not total 40. Are there studies both assessing availability and accessibility&usage?

Is it possible to indicate which studies corresponde to each dimension?

Line 216-217: Please revise the sentence “Overall, there was mixed evidence a statistically significant relationship between exposures and outcomes”

Line 219: “but majority did not find a statistically significant association ”. How can the authors refer majority with six studies? The majority usually is used above 75%

Table 1: The meaning of the abbreviations should appear in footnote - NDVI, HR, …

DISCUSSION

With such organization and detail in the previous section, I was a little bit disappointed about the simplicity of the Discussion. I think that the authors should better confront their findings with existing evidence and try to give possible explanations for some of the results. For example, if the results from cross-sectional studies show a positive association between exposure and health outcomes, how come this association is lost in longitudinal studies?

Line 337: Can the authors explain why “This is important for informing intervention design and policy decision making. ”?

SUPPLEMENTARY MATERIAL

The title of the Supplementary material 2 is: Supplementary Material 1: Data Extraction Form Adapted from Cochrane 

I think that where is 1 should be 2.

Please also include the Cochrane reference in this document

I don’t understand the numerical sequence in column 1 of the Supplementary material 4

Supplementary material 5

The column “Demonstration outcome of interest not present at “ is not totally visible

Author Response

We'd like to thank the reviewer for the thorough, constructive and thoughtful feedback on our systematic review. We've taken into consideration every point and revised the manuscript accordingly.

Please see attachment for further details.

Reviewer 2 Report

The paper intends to examine the relationship between exposure to blue and green spaces with mental health conditions, NCDS, health-relation behaviours and other health states. Moreover, the author intends to examine the relationship in longitudinal studies and for the risk of developing multiple chronic conditions. To answer these, the authors conducted systematic review of 44 studies.

The paper is well-developed with clear objective, structure and research design. However, I would suggest the authors to clarify the quality appraisal assessment process of the studies. In the Supplementary material 3 the authors have assessed each paper across three measures: selection, comparability and outcome assessment. What is not clear is how the authors have come up with the overall rating of good, fair and bad.

Author Response

We thank the reviewer for their kind feedback on our systematic review. We're pleased our paper comes across as well-developed and structured. 

We've clarified our quality assessment in the main body text and the supplementary material. 

Please see the attachment with more detailed descriptions and responses. 

Round 2

Reviewer 1 Report

The authors have now properly addressed all my previous comments. I suggest that the manuscript should be accepted in its present form